# Hepatitis E Vaccination Preferences and Willingness-to-Pay Among Residents: A Discrete Choice Experiment Analysis

**DOI:** 10.3390/vaccines13090906

**Published:** 2025-08-27

**Authors:** Yuanqiong Chen, Chao Zhang, Zhuoru Zou, Weijun Hu, Dan Zhang, Sidi Zhao, Shaobai Zhang, Qian Wu, Lei Zhang

**Affiliations:** 1School of Public Health, Xi’an Jiaotong University Health Science Center, Xi’an 710061, China; yqchen099@stu.xjtu.edu.cn (Y.C.); zrzou123@xjtu.edu.cn (Z.Z.); sidizhao@stu.xjtu.edu.cn (S.Z.); 2Shaanxi Provincial Centre for Disease Control and Prevention, Xi’an 710054, China; zhangchao@sxcdc.com (C.Z.); huweijun@sxcdc.com (W.H.); zhangdan@sxcdc.com (D.Z.); 3Shaanxi Provincial Bureau of Disease Prevention and Control, Xi’an 710043, China; shaobai@sxcdc.com; 4Phase I Clinical Trial Research Ward, The Second Affiliated Hospital of Xi’an Jiaotong University, Xi’an 710004, China; 5China-Australia Joint Research Center for Infectious Diseases, School of Public Health, Xi’an Jiaotong University Health Science Center, Xi’an 710061, China; 6Artificial Intelligence and Modelling in Epidemiology Program, Melbourne Sexual Health Centre, Alfred Health, Melbourne 3053, Australia; 7School of Translational Medicine, Faculty of Medicine, Nursing and Health Sciences, Monash University, Melbourne 3053, Australia

**Keywords:** hepatitis E vaccine, discrete choice experiment, preference heterogeneity, willingness-to-pay

## Abstract

**Objectives**: Hepatitis E virus (HEV) infection is associated with severe hepatitis and high mortality rates, yet vaccination coverage remains suboptimal. Investigating public preferences for HEV vaccination is critical for developing targeted prevention strategies. This study employed a discrete choice experiment (DCE) to quantify attribute preferences and willingness-to-pay (WTP) for HEV vaccination among Chinese residents (in Shaanxi Province, for example), aiming to inform evidence-based immunization policy optimization. **Methods**: A cross-sectional survey recruited 3300 participants using stratified random sampling. The vaccine attributes—protective efficacy, duration of protection, and out-of-pocket cost—were identified using a systematic literature review and expert consultation. A comparative analysis of preference characteristics was conducted using conditional logit (Model 1) and mixed logit (Model 2) regression models. Population heterogeneity in vaccination preferences was further analyzed using the conditional logit framework, with marginal WTP estimated using parameter coefficients. **Results**: Among 3199 valid responses, duration of protection (Model 2: 10-years; *β* = 0.456, *p* < 0.001) and out-of-pocket cost (Model 2: 2000–3000 CNY; *β* = −0.179, *p* < 0.001) significantly influenced vaccination decisions. Preference heterogeneity was observed: women of childbearing age prioritized longer protection (10 years; *β* = 0.677, *p* < 0.001) and were sensitive to the cost of 1000–2000 CNY (*β* = 0.169, *p* = 0.011), while urban residents valued extended protection more than rural counterparts. **Conclusions**: Protection duration emerged as the primary determinant of HEV vaccination preference. Policy recommendations include implementing tiered pricing strategies and targeted health education campaigns emphasizing long-term protection benefits to enhance vaccine uptake and affordability.

## 1. Introduction

Viral hepatitis, as one of the critical global public health challenges, directly impacts the achievement of the United Nations Sustainable Development Goals. Hepatitis E virus (HEV) infection, recognized as the leading cause of acute sporadic viral hepatitis worldwide, has persistently suffered from severe underestimation of its disease burden. HEV spreads via fecal–oral transmission, bloodborne routes, and zoonotic pathways, exhibiting endemic prevalence in low-resource settings and low-to-middle-income countries [1]. Annually, HEV accounts for approximately 20 million infections worldwide, including 3.3 million symptomatic cases and 44,000 fatalities [2]. Despite the generally self-limiting course and favorable prognosis observed in most immunocompetent infected individuals [3], HEV infection can lead to catastrophic outcomes in specific high-risk populations. Pregnant women, chronic liver disease patients, and immunocompromised individuals face elevated risks of fulminant hepatitis, hepatic failure, and mortality. Notably, third-trimester HEV infections incur case fatality rates of 20–25% [4], which is much higher than in the general population. It is also significantly associated with adverse pregnancy outcomes such as preterm birth, stillbirth, and vertical transmission. In China, the disease burden of hepatitis E is also not negligible. Since 2012, the incidence rate of hepatitis E in China (2.02 per 100,000) has surpassed that of hepatitis A (1.81 per 100,000) [5]. The substantial burden of this disease is further exacerbated by high hospitalization costs and the lack of specific therapeutic agents [6]. Shaanxi Province serves as a crucial region in Northwest China due to its high representativeness in terms of China’s geography, demographics, and socioeconomic characteristics. Geographically spanning both northern and southern China, Shaanxi encompasses the urbanization of the Guanzhong Plain, the features of the Loess Plateau in northern Shaanxi, and the mountainous terrain of the Qinba Mountains in southern Shaanxi. Its population structure, economic development level, distribution of healthcare resources, and ethnic composition (predominantly Han Chinese with some minority groups) collectively make it a microcosm of China. The global seroprevalence of anti-HEV IgG is 12.47% [7], indicating widespread prior infection. A 2017 seroepidemiological survey in Shaanxi Province of China revealed a significantly higher prevalence of 15.40% among the general population [8], classifying it as a high-endemic region for hepatitis E. However, underdiagnosis due to suboptimal HEV detection sensitivity likely underestimates disease burden [9,10], underscoring the urgency for enhanced vaccination in high-risk populations and foodborne transmission control.

No HEV-specific antiviral therapies have been globally approved, leaving supportive care as the primary clinical intervention [9,11]. Vaccination thus emerges as the cornerstone strategy for HEV containment. Notably, China has achieved a groundbreaking milestone in this field with the domestically developed recombinant hepatitis E vaccine (expressed in Escherichia coli system) pioneered by Xiamen University and collaborators, which received initial regulatory approval in 2012 for individuals aged 16 years and above. Pivotal phase III clinical trial data and extended follow-up studies demonstrated 86.8% protective efficacy against symptomatic hepatitis E over 4.5 years post-vaccination, accompanied by robust immunogenicity evidenced by a seroconversion rate exceeding 98% following the complete vaccination schedule [12]. This vaccine represents the world’s first and currently sole licensed prophylactic HEV vaccine, marking a significant milestone in global public health. WHO’s 2015 position paper endorses its protective efficacy in healthy Chinese populations aged 16–65 and advocates context-specific immunization strategies [13]. Globally, systematic vaccination campaigns achieved 58% full vaccination coverage in South Sudan’s high-endemicity zones [14] and Bentiu region has reached 95% coverage in displaced persons camps (27,000 cases) by April 2022 [15]. In contrast, China reports only 8.81% vaccine uptake in Shanghai [16], with regional data elsewhere remaining unreported. This mismatch between immunological protection and disease burden is most pronounced among women of childbearing age, a critical target group given their elevated mortality risk during late pregnancy. Despite this urgency, HEV vaccination uptake remains suboptimal due to multifaceted barriers including vaccine safety concerns, fragmented health communication, and reproductive health considerations.

Understanding individual preferences is paramount for optimizing resource allocation and enhancing intervention uptake in public health policymaking and health economic evaluations. This is particularly pertinent to vaccination decisions, where success critically depends on individuals’ voluntary engagement and active participation.

Discrete choice experiments (DCEs) have emerged as a robust tool for vaccine preference research, simulating real-world decision-making contexts. Spanish studies reveal parental prioritization of vaccine safety versus clinicians’ efficacy focus [17]. New Zealand research shows that the risk of adverse reactions to vaccines is the attribute that the general public cares about most, but adolescents are more concerned about the burden of disease [18]. Guizhou’s DCE application highlights parental sensitivity to pneumococcal vaccine pricing and accessibility [19]. Over 45 DCE studies in the past five years have mapped preferences for influenza and HPV vaccines across diverse populations [20]. These studies not only validate the robustness and cross-cultural applicability of DCE, but also accumulate rich practical experience. However, HEV vaccine preference research remains conspicuously absent. Given China’s unique position as the sole country with a licensed HEV vaccine globally, coupled with the pronounced disparity between high disease burden and low vaccination uptake, this research gap becomes particularly critical to address. Consequently, this study focuses on Shaanxi Province—a representative high-endemic region for hepatitis E in China—employing a rigorous discrete choice experiment (DCE) methodology integrated with advanced econometric models (e.g., mixed logit models) to propose a systematic analysis of the preference characteristics of Shaanxi residents for HEV vaccination, quantify the impact of vaccine attributes on decision making, estimate the willingness-to-pay (WTP), and dissect subgroup heterogeneity. The findings are anticipated to provide pivotal scientific evidence for developing precision-targeted and effective HEV vaccination promotion strategies and immunization policies in Shaanxi Province and nationwide. By identifying key attributes and population characteristics that drive or hinder vaccination uptake, the results will directly inform policymakers to optimize vaccine attributes (e.g., enhancing communication strategies to emphasize safety evidence), refine pricing and financing mechanisms, improve accessibility and convenience of vaccination services, and design tailored health education and social mobilization programs for specific subpopulations (particularly high-risk groups with low willingness such as women of childbearing age).

Ultimately, this research will contribute substantially to increasing HEV vaccine coverage, reducing morbidity and mortality among high-risk populations, and alleviating the disease burden of this long-neglected major public health challenge.

## 2. Materials and Methods

### 2.1. Participant Recruitment

The study utilized data from Shaanxi Province’s 2017 hepatitis B seroepidemiological survey to select 10 districts/counties across 8 cities (Yan’an City, Yulin City, Xi’an City, Baojing City, Xianyang City, Weinan City, Shangluo City, and Ankang City), spanning three geographical regions (Guanzhong, Northern Shaanxi, and Southern Shaanxi), as detailed in Figure 1. Sampling sites encompassed urban communities and rural townships with varying urbanization levels, socioeconomic structures, healthcare resource allocations, and population densities to comprehensively represent HEV vaccination demand.

(1) Recruitment Protocol

This study employed a stratified random sampling approach to select representative residents at the county level. Initially, within each of the 10 pre-selected counties/districts, three administrative units (i.e., administrative villages or urban communities) were randomly chosen as primary sampling sites using simple random sampling. Subsequently, within each selected village/community, eligible permanent resident registries (household registration or residency records) were obtained from local village/neighborhood committees or community health centers to serve as sampling frames. Predetermined numbers of target households were then systematically selected from these registries using computer-generated random number tables.

(2) Inclusion Criteria

Research subjects must meet all of the following conditions:(a)Permanent residency: Continuous residence ≥6 months in the sampled administrative village/community as of the survey date, irrespective of household registration status;(b)Age range: 15 to 66 years inclusive, verified through government-issued identification or household registration documents;(c)Informed consent: After trained investigators comprehensively explained the study purpose, procedures, potential benefits/risks (primarily time commitment and privacy safeguards), voluntary participation, right to withdraw unconditionally, and confidentiality measures in a clear, non-suggestive manner, participants provided written informed consent. For non-literate individuals, impartial witnesses attested to the complete disclosure process before participants affixed their thumbprint, with both investigator and witness countersigning.

(3) Exclusion Criteria

Those who meet any of the following conditions will be excluded:(a)Comprehension and communication barriers: Individuals with severe hearing, linguistic, or cognitive impairments (e.g., intellectual disability, advanced dementia) were excluded if investigators assessed them as incapable of comprehending survey content or engaging in basic communication during the consent process;(b)Psychiatric disorders: Clinically diagnosed patients with severe mental illnesses (e.g., schizophrenia, bipolar disorder in acutely symptomatic phase) exhibiting unstable conditions that could impair rational decision making or study understanding.

### 2.2. Study Design

#### 2.2.1. Attribute and Level Identification

A systematic literature review [21,22,23] identified vaccine efficacy, cost, duration of protection, and adverse event risk as primary determinants of public vaccination preferences. Five candidate attributes were initially proposed based on contextual analysis of 10 Shaanxi counties: protective efficacy, duration of protection, severe adverse event rate, vaccination site, and out-of-pocket cost. To ensure the attributes and levels employed in the DCE authentically, effectively, relevantly, and actionably reflect the actual preferences and decision trade-offs of the target population in Shaanxi Province, the research team implemented a rigorous two-phase screening and validation protocol:

Phase 1: Expert Panel Review

A six-member advisory panel was convened, comprising authorities with extensive expertise in infectious disease control, vaccinology, health economics, epidemiology, public health policy, and primary healthcare management. Panelists were recruited from provincial Centers for Disease Control and Prevention (CDC), university schools of public health, and primary healthcare institutions within sampled counties to ensure diversified perspectives. Focus group discussions were conducted to critically evaluate the scientific validity, practical relevance, and potential implementation challenges of proposed attributes.

Regarding the “serious adverse event incidence rate” attribute, the panel presented the following key arguments during in-depth deliberation:(a)Post-marketing surveillance data from long-term, large-scale HEV vaccination programs (e.g., the China-licensed vaccine [13]) demonstrate an incidence rate orders of magnitude lower than common vaccines.(b)China’s established and rigorous Adverse Events Following Immunization (AEFI) surveillance system ensures robust safety monitoring, maintaining relatively high public trust.

The consensus indicated that retaining this attribute would likely fail to discriminate preferences while unnecessarily inflating questionnaire complexity and respondent cognitive burden. Consequently, the panel strongly recommended its exclusion—a recommendation subsequently implemented.

Phase 2: Pilot Survey and Cognitive Testing

The retained attributes (efficacy, duration, location, and out-of-pocket cost) and their levels underwent empirical testing within the target population to assess comprehensibility, realism, acceptability, and discriminatory power. One representative county (encompassing urban and rural settings) was purposefully selected from the 10 sampled counties in Shaanxi Province. Approximately 100 eligible residents were recruited using convenience sampling supplemented by quota sampling to achieve balanced age, gender, and urban–rural distribution.

Participants first completed the draft questionnaire containing DCE choice sets. Subsequently, trained researchers conducted one-on-one cognitive interviews. Feedback indicated the following:(a)Most participants clearly understood “efficacy,” “duration,” and “cost” attributes.(b)Regarding “vaccination location,” respondents widely reported minimal perceived differences in convenience (e.g., distance, time cost) across healthcare tiers (village clinics, township health centers, county hospitals) due to Shaanxi’s well-established primary healthcare network. Typical responses included the following: “Similar convenience regardless of site” or “Routine vaccination occurs at nearest local clinics.”

This finding suggested limited discriminatory utility for the location attribute. Consequently, the decision was made to exclude “vaccination location” to streamline the questionnaire, focus on core decision drivers, and enhance data validity and reliability.

Through a rigorous tripartite screening protocol encompassing a literature review, expert panel review, and target population pilot validation, three key attributes that best capture the fundamental determinants of hepatitis E vaccination decision making among Shaanxi residents were ultimately identified for the formal discrete choice experiment (Appendix A).

#### 2.2.2. DCE Framework

A full factorial design generated 18 theoretical vaccine profiles (2 efficacy levels × 3 duration levels × 3 cost levels).

Given that a full factorial design comprising 18 vaccine profiles would generate 153 combinations if employing an initial choice set framework with two randomly paired profiles sampled without replacement—significantly increasing respondent burden—this study adopted a D-optimal design strategy to balance methodological rigor and respondent experience. Specifically, the %ChoicEff macro in SAS 9.4 software was utilized to screen 20 statistically optimal choice sets from the full factorial design space, achieving a D-efficiency score of 14.30. Each choice set included two vaccine profiles with differing attribute levels and an “opt-out” option, ensuring pronounced inter-profile attribute variability while controlling cognitive burden. Example choice sets are provided in Appendix A.

To enhance data collection efficiency, the 20 choice sets were partitioned into four randomized blocks (5 sets per block) using a randomized block design.

#### 2.2.3. Questionnaire Development and Piloting

A structured questionnaire was developed through literature synthesis, semi-structured interviews, and expert consultations. Closed-ended questions were prioritized for analytical efficiency. The questionnaire comprised:Instructions: Study background, objectives, and completion guidelines.Demographics: Age, gender, education, income, etc.Health status: Chronic liver disease history, prior vaccination experience.Vaccination intent: Willingness to receive HEV vaccination.DCE section: Presented choice sets requiring participants to select preferred alternatives.

A pilot study involving 100 participants in Huayin County of Shaanxi Province assessed attribute interpretability and questionnaire feasibility. Attributes and levels were refined based on pretest outcomes, while ambiguous or inappropriate items were revised to enhance logical coherence and respondent clarity.

### 2.3. Sample Size Estimation

The precision and stability of parameter estimation in DCE, a stated-preference methodology grounded in random utility theory, are highly dependent on sample size. This study integrated two mainstream approaches for sample size determination.

First, following the empirical rule proposed by Johnson and Orme [24]—widely adopted in DCE research—the minimum sample size was calculated using the formula:(1)$$n>\frac{500\;\times \;c}{t\;\times \;a}$$
where *t* denotes the number of choice sets (5), *a* the alternatives per choice set (2), and *c* the maximum attribute level count (3). Substituting these parameters yielded a minimum requirement of *n* > 150, but this only meets the most basic requirements for model parameter estimation. However, it is generally insufficient for complex analyses or detecting small effect sizes between attribute levels with high precision.

Given that empirical rules provide only a baseline threshold, this study further referenced empirical DCE studies and scholarly recommendations. As noted by Sargazi et al. [25] and Hofman et al. [26], multiple simulation analyses and practical DCE implementations suggest that sample sizes of 300–400 participants (or generally >300) provide adequate statistical power for robust and reliable parameter estimation. This approach prioritizes analytical feasibility and result stability, complementing the lower bound derived from empirical formulas.

Based on the above dual criteria, this study adopted a stratified sampling strategy:(a)Basic sample: 300 residents were sampled from each of the 10 sample counties, for a total of 3000 people.(b)Professional sample: An additional 300 healthcare workers were selected.

The total sample size is 3300 individuals, which significantly exceeds the minimum sample requirement (150). This design meets the needs for obtaining robust parameter estimates, exploring potential preference heterogeneity, conducting subgroup comparison analyses, and maintaining high statistical test power.

### 2.4. Data Collection

This study collected data through an electronic questionnaire system. Survey points were set up in vaccination clinics at county-level hospitals, township health centers, and community health service centers. Participants scanned a QR code or clicked on a dedicated link to access the online platform and complete the questionnaire. Trained investigators provided standardized instructions on study objectives and completion protocols. To ensure questionnaire quality, questionnaires were considered invalid under the following criteria: (1) logical inconsistencies were identified if respondents refused a clearly dominant alternative within any DCE choice set (e.g., selecting “80–90% protective efficacy + 10-year duration + 2000–3000 CNY” while rejecting “90–100% protective efficacy + 30-year duration + 0–1000 CNY”); (2) critical data omissions were deemed invalid if respondents selected “Opt-out” for all choice sets. Of 3300 distributed questionnaires, 3199 valid responses were retained after excluding invalid submissions (logical inconsistencies or critical data omissions), yielding a 96.94% valid response rate.

### 2.5. Data Analysis Methods

#### 2.5.1. Descriptive Statistics

Statistical analyses were conducted using Stata 17.0. Continuous variables were summarized as mean (standard deviation, SD), while categorical variables were reported as frequency (*n*) and percentage (%).

#### 2.5.2. Mixed Logit Model

The mixed logit model was employed as the core analytical method, incorporating random parameters to capture individual preference heterogeneity. This model was selected based on its significantly superior goodness-of-fit compared to the conditional logit model, as evidenced by lower Akaike Information Criterion (AIC)/Bayesian Information Criterion (BIC) values.

Random parameters were specified as follows:(a)Random parameters: protective efficacy, duration of protection, and out-of-pocket cost;(b)Distributional assumption: All random parameters were assumed to follow a normal distribution;(c)Correlation structure: Random parameters were specified as mutually independent, resulting in a diagonal covariance matrix (implemented by omitting the “corr” option in Stata17.0’s “mixlogit” command);(d)Variance estimation: The model estimated the standard deviation and statistical significance of each random parameter.

The model was fitted using simulated maximum likelihood estimation (SMLE). Estimation efficiency was enhanced through 500 simulation repetitions, achieved by setting “nrep(500)’’ in Stata 17.0. Covariates included in the model are detailed in Table 1. Subsequently, interaction terms between covariates and subgroup variables were introduced to identify between-group differences. Statistical significance (*p* < 0.05) of interaction coefficients and their 95% confidence intervals (CI) served as the criteria for interpretation.

#### 2.5.3. WTP Analysis

Marginal WTP was calculated via the mixed logit model using the formula:(2)$$WTP_{x}=-\frac{\beta_{x}}{\beta_{cost}}$$
where *x* denotes vaccine attributes, *β_x_* the regression coefficient for attribute *x*, and *β_cos_t* the coefficient for out-of-pocket cost. Furthermore, this formula requires the cost variable to exhibit linear (continuous) effects within the utility function. Given that the cost data in this study were collected in categorical interval format, they were converted into a continuous variable during subsequent analyses to approximate compliance with this assumption. Specifically, midpoint imputation was applied for conversion (e.g., the interval “A–B” was assigned the value (A + B)/2).

Mixed logit models were used in all analyses exploring subgroup preference heterogeneity or residential heterogeneity in vaccine preferences for the hepatitis E vaccine.

### 2.6. Ethics

The study was conducted in accordance with the Declaration of Helsinki and approved by the Ethics Committee of the First Affiliated Hospital of Xi’an Jiaotong University (protocol code XJTU1AF2025LSYY-021). All participant information was maintained under strict confidentiality, with questionnaires identified solely by research codes. The linkage table connecting personal identifiers to research data was encrypted and stored separately. Participants retained the unconditional right to withdraw from the study at any stage without incurring adverse consequences. Findings will be reported exclusively in aggregated form, ensuring no individual-level information is disclosed.

## 3. Results

### 3.1. Participant Characteristics

The study cohort comprised 3199 participants with a mean age of 38.11 ± 13.83 years. Females accounted for 51.58% of the sample, with 79.27% residing in rural areas. Educational attainment was predominantly at the junior high school level (31.20%). Annual income distribution revealed 34.82% earning between 10,000 and 30,000 CNY. Occupational profiles indicated industrial/commercial workers and production practitioners as the largest group (65.74%), followed by students (14.07%), with retirees representing the smallest proportion (1.25%). Hepatitis-related medical history was rare, with 2.00% reporting familial hepatitis history and 0.53% having personal hepatitis diagnoses. Detailed demographic characteristics are presented in Table 2.

### 3.2. Attribute-Specific Influences on Vaccination Decisions

The goodness-of-fit of the conditional logit model (Model 1) and mixed logit model (Model 2) is presented in Table 3. Duration of protection emerged as the dominant attribute, particularly favoring 10-year coverage (Model 1: *β* = 0.407, *p* < 0.001; Model 2: *β* = 0.456, *p* < 0.001). Out-of-pocket cost exhibited consistent negative associations across models. Protective efficacy showed statistically non-significant effects in both models (Table 4 and Table 5).

### 3.3. WTP Analysis for Hepatitis E Vaccine

WTP estimates revealed limited economic valuation for efficacy improvements (80–90% to 90–100%: WTP = 0.102 CNY, *p* = 0.802). Conversely, extended duration of protection (5 to 10 years) generated substantial WTP increments (7.077 CNY, *p* < 0.001), indicating strong preference for prolonged coverage (Table 6). This result is calculated based on the converted continuous cost variable.

### 3.4. Subgroup Preference Heterogeneity

Both population groups exhibited a highly significant preference for longer protection durations (10 years/30 years vs. 5 years; *p* < 0.001), indicating that extended protection duration is a core attribute for improving vaccination willingness (Figure 2 and Figure 3). In contrast, enhanced efficacy (90–100% vs. 80–90%) showed no significant impact in either group (healthcare workers: *p* = 0.101; women of childbearing age: *p* = 0.928).

Among healthcare workers, increased cost (2000–3000 CNY vs. 0–1000 CNY) demonstrated no significant negative effect (*p* = 0.111). For women of childbearing age, greater acceptance was observed for moderate costs (1000–2000 CNY; *β* = 0.169, *p* = 0.011), while no significant resistance emerged toward higher costs (2000–3000 CNY: *β* = −0.155, *p* = 0.111). This result is based on calculations using the mixed logit model.

### 3.5. Residential Heterogeneity in Vaccine Preferences

Residents from both rural and urban areas showed no preference for paying 1000–2000 CNY for vaccination (*p* = 0.345 vs. *p* = 0.644, respectively). Compared to short-term protection (5 years), both groups exhibited a significant preference for longer protection durations (10 years/30 years; *p* < 0.001 for both). Regarding enhanced efficacy (90–100% vs. 80–90%), rural residents showed no response (*β* = −0.010, *p* = 0.739), while urban residents demonstrated a non-significant positive trend (*β* = 0.106, *p* = 0.168) (Table 7). This result is based on calculations using the mixed logit model.

## 4. Discussion

The duration of protection is a core factor influencing the preference for hepatitis E vaccines. The study reveals that protection periods of 10 and 30 years are particularly favored by residents, with a pronounced preference for the 10-year duration. This tendency might be related to the cost reduction associated with long-lasting vaccines, a pattern also confirmed in other vaccine studies: parents were willing to pay more for a protection period exceeding 10 years for pneumococcal vaccines [19], protection duration was the primary concern in the Australian meningitis vaccine survey [27], and both Chinese and American residents showed a preference for long-lasting vaccines in the choice of COVID-19 vaccines [28]. From a public health perspective, extending the vaccine protection period has dual benefits. On one hand, long-lasting vaccines can increase vaccination willingness by reducing the frequency of vaccinations; on the other hand, enduring protective effects can enhance public trust in vaccine efficacy, thereby creating a virtuous cycle of “high trust-high vaccination rates.”

Optimizing the out-of-pocket mechanism for hepatitis E vaccination to enhance long-term protection awareness and increase willingness-to-pay. This study reveals distinct characteristics of group sensitivity to out-of-pocket costs associated with hepatitis E vaccination. Economic burdens significantly impact vaccination decisions, with low-income groups being particularly sensitive. This finding is consistent with numerous prior studies [29,30,31]. Data indicate that 2000 CNY may represent a critical psychological threshold. When the cost is below this level, the essential demand for the vaccine or the positive perception of other attributes (such as long-term protection) effectively reduces price resistance; however, once this threshold is surpassed, sensitivity to price markedly increases. This suggests that controlling costs within a reasonable range can effectively eliminate economic barriers. In other studies, the positive influence of vaccine efficacy on willingness-to-pay has been corroborated. A study conducted in the United States [32] demonstrated that when vaccine efficacy rises from 50% to 95%, the willingness-to-pay increases by 21.7%, with similar trends observed in COVID-19 vaccine data [33]. This study does not observe similar results. The potential reason may be that the increase in vaccine efficacy (from 80–90% to 90–100%) fails to significantly enhance the respondents’ willingness-to-pay. The relationship between duration of protection and willingness-to-pay is nonlinear: the price premium for extending protection from 5 years to 10 years is higher than that for extending it from 10 years to 30 years. This may relate to health planning cycles (such as risk windows at specific age intervals) and expectations of technological advancements. The risk aversion associated with medium- to long-term immunity directly translates into payment incentives; however, excessively long protection periods may lead to diminishing marginal utility due to a disconnect from actual needs.

Understanding the vaccination needs for hepatitis E among women of childbearing age is vital for enhancing health education and effectively promoting tailored vaccination programs. This demographic plays a crucial role in hepatitis E prevention, and the existing literature suggests that vaccination behaviors are significantly influenced by individual characteristics such as income and education levels [34,35,36]. However, there is a lack of explicit mention in the literature regarding their preferences for vaccine efficacy and protection duration. This study identifies important insights regarding the vaccination demand among women of childbearing age: this group exhibits a stronger sensitivity to vaccine protection duration (10 years or 30 years) and out-of-pocket cost than to the efficacy range (80–90% vs. 90–100%). However, when costs increase, especially when prices exceed 2000 CNY, vaccination willingness decreases significantly (*β* = −0.008, *p* = 0.886). Such preferences are closely linked to their unique reproductive health needs; opting for long-lasting vaccines can reduce the frequency of required vaccinations and provide a sustained immune barrier for their families. Additionally, in the context of rising medical expenses and childcare costs, economic accessibility emerges as a critical factor in their decision-making process. Based on these findings, a “demand-driven” intervention strategy is recommended: health education should emphasize the long-term protective benefits of vaccines, and precise recommendations are implemented through counseling by medical personnel to communicate birth plans and economic status.

It is important to provide precise and personalized vaccination programs informed by heterogeneity in population preferences. While previous studies [37,38,39,40] primarily examined disparities in vaccination rates between urban and rural populations, this research delves deeper into the differing preferences for vaccine characteristics among these groups. The data show that there is consensus among urban and rural groups on two aspects: the value of long-term protection is generally recognized and both are sensitive to out-of-pocket costs. Based on these findings, differentiated strategies are recommended. Specifically, mid-term protective vaccines (5–10 years) should be promoted in rural areas, supported by community health workers leading public education on “illustrating vaccine efficacy”; meanwhile, urban areas should prioritize long-lasting vaccines and implement a tiered medical insurance reimbursement structure to alleviate financial burdens, thus enhancing vaccination efficiency and reducing health equity gaps.

A stratified sampling strategy encompassing 10 urban and rural districts/counties with heterogeneous socioeconomic profiles in Shaanxi Province ensured geographical representativeness. In the experimental design phase, a full factorial design and D-optimal design were utilized to select the 20 choice sets with optimal statistical efficiency. Block design was applied to balance the complexity of the options, ensuring analytical power while reducing cognitive load on respondents. In terms of quality control, the research team conducted a pre-survey in Huayin County to refine the attribute settings of the questionnaire, providing standardized training to enumerators to ensure response consistency. Data collection adopted a mixed-mode approach, integrating online and offline methods, where the mandatory options of the electronic questionnaire was combined with on-site guidance to effectively avoid invalid responses and missing key information. Dual-entry verification and quality audits enhanced data integrity and accuracy.

However, this study presents several limitations. Methodologically, the DCE requires respondents to make trade-off decisions across multiple attributes, which may pose barriers to understanding for individuals with lower educational backgrounds. Although the questionnaire design was improved through a pre-survey, completely eliminating the risk of cognitive bias remains challenging, particularly regarding potential biases in decision making. Geographically, while the sample encompasses various regions of Shaanxi Province, caution should be exercised when extrapolating findings to other settings due to the specific characteristics of healthcare resource allocation in these areas.

A limitation of this study stems from the approach to cost data collection. Cost information was obtained as predefined intervals rather than precise continuous values. Although these intervals were converted into a continuous variable (using midpoint imputation) for WTP calculations during analysis, this transformation relies on assumptions regarding the cost distribution within intervals (presumed to be uniform). Consequently, the reported WTP values should be interpreted as approximations of the true continuous marginal WTP. Future research designs should prioritize collecting more precise cost data (e.g., actual payment amounts or finer cost levels) or employ specialized modeling techniques for interval data (e.g., interval regression) to enhance the precision of WTP estimates. Nevertheless, the WTP estimates presented herein retain significant value for comparing the relative importance of attributes and informing management decisions.

## 5. Conclusions

This study identifies the duration of protection as the key influencing factor in HEV vaccination decisions. The public shows a strong preference for vaccines that provide long-term protection for 10 years, while improvements in protective efficacy do not significantly affect their choices. Vaccination willingness decreases significantly when out-of-pocket costs exceed 2000 CNY, indicating pronounced economic sensitivity. Women of childbearing age are particularly focused on long-term protection, while both urban and rural residents acknowledge the importance of protection duration. Policy recommendations include the following: (1) prioritizing vaccines with ≥10-year protection; (2) tiered pricing strategies to mitigate financial barriers; (3) demographic-specific immunization programs.

Future research should consider expanding the sample size and geographic coverage, exploring additional attributes and their levels, employing advanced modeling techniques (e.g., latent class analysis) to refine preference heterogeneity mapping, and integrating long-term follow-up surveys to provide a comprehensive assessment of vaccination preferences and their influencing factors.

## Figures and Tables

**Figure 1 vaccines-13-00906-f001:**
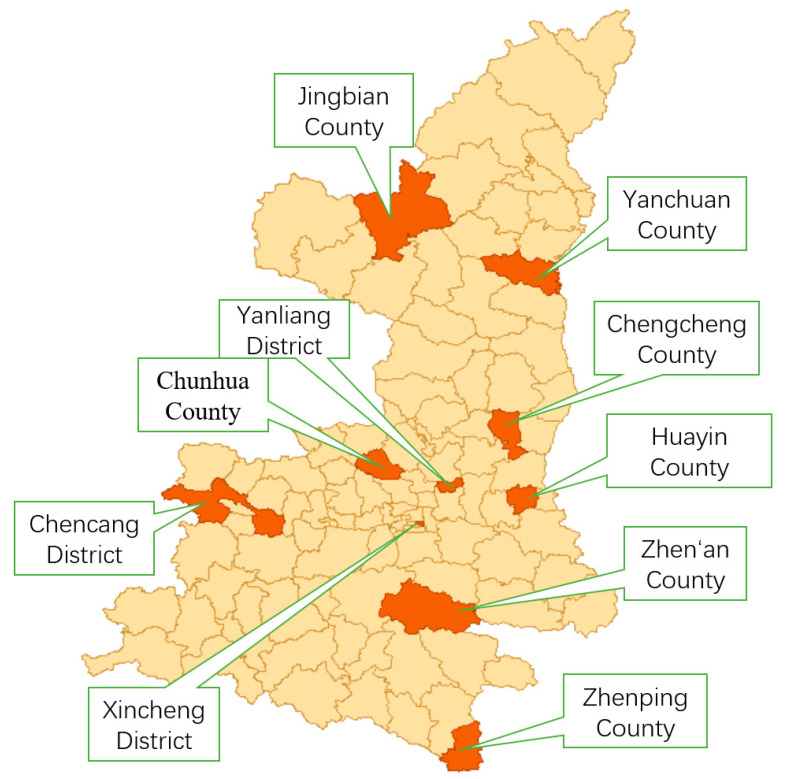
Geographical distribution map of sampling districts and counties in Shaanxi Province.

**Figure 2 vaccines-13-00906-f002:**
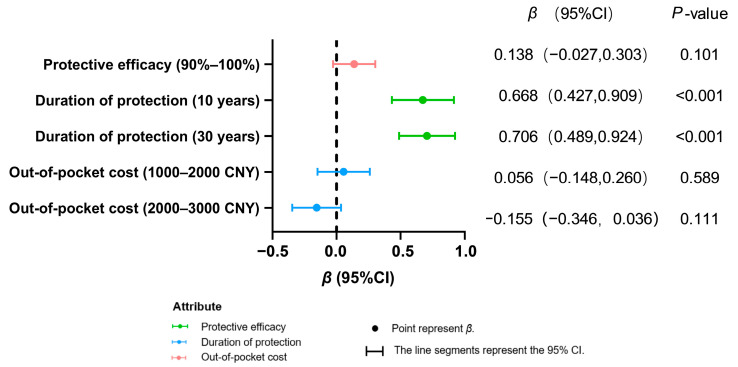
Preference analysis of health workers’ decision making on hepatitis E vaccination.

**Figure 3 vaccines-13-00906-f003:**
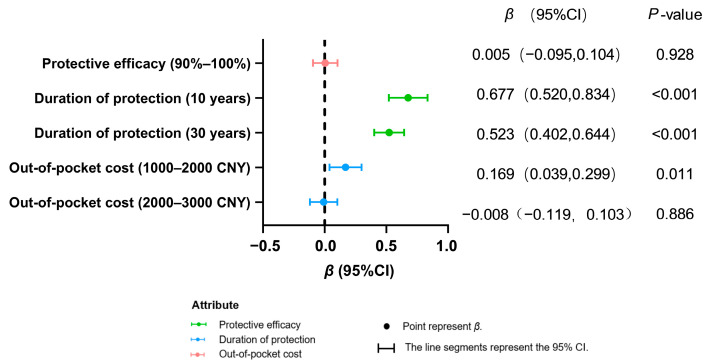
Preference analysis of women of childbearing age for decision making on hepatitis E vaccination.

**Table 1 vaccines-13-00906-t001:** Included variables.

Variable Category	Variable Name	Coding/Levels	Rationale for Inclusion
Demographic Characteristics	Gender	1 = Male 2 = Female	Control for gender-related preference differences
	Occupation	1 = Health workers 2 = Non-health workers	Control for professional background influence
	Education Level	1 = Below college 2 = Over College	Control for knowledge disparity
Economic Factors	Annual Income	1 = Below 50,000 CNY 2 = Over 50,000 CNY	Control for payment capacity differences
Vaccine Attributes	Protective Efficacy	1 = 80–90% 2 = 90–100%	Key effectiveness evaluation metric
	Duration of Protection	1 = 5 years 2 = 10 years 3 = 30 years	Key durability assessment metric
	Out-of-Pocket Cost	1 = 0–1000 CNY 2 = 1000–2000 CNY 3 = 2000–3000 CNY	Key cost evaluation metric

CNY: Chinese Yuan.

**Table 2 vaccines-13-00906-t002:** Demographic characteristics of subjects.

Variable	Cluster	Value
Age (years)	Mean (SD)	38.11(13.83)
Gender (%)	Male	1549(48.42)
	Female	1650(51.58)
Region (%)	Rural	2536(79.27)
	Urban	663(20.73)
Educational level (%)	Primary and lower	466(14.57)
	Junior high school	998(31.20)
	High school/technical secondary school	743(23.23)
	University/professional training college	965(30.17)
	Postgraduate and above	27(0.84)
Annual income (%)	<10,000 CNY	625(19.54)
	10,000–30,000 CNY	1114(34.82)
	30,000–50,000 CNY	905(28.29)
	50,000–100,000 CNY	398(12.44)
	>100,000 CNY	157(4.91)
Occupation (%)	Health workers	324(10.13)
	Students	450(14.07)
	Educational and public service practitioners	282(8.82)
	Industrial/commercial workers and production practitioners	2103(65.74)
	Retirees	40(1.25)
Familial hepatitis history (%)	Yes	64(2.00)
	No	2533(79.18)
	Not clear	602(18.82)
Personal hepatitis history (%)	Yes	17(0.53)
	No	2679(83.74)
	Not clear	503(15.72)

SD: standard deviation.

**Table 3 vaccines-13-00906-t003:** Model goodness-of-fit.

Model	Log-Likelihood	AIC	BIC
Conditional logit model (Model 1)	−6217.917	12,445.83	12,484.90
Mixed logit model (Model 2)	−6158.312	12,370.62	12,581.56

**Table 4 vaccines-13-00906-t004:** Influence of vaccine attributes on vaccine choice preferences based on the conditional logit model (Model 1).

Attributes and Levels	Coefficient (SE)	*p*-Value	95% CI
Protective efficacy			
80–90% (Ref)			
90–100%	0.006 (0.021)	0.768	(−0.036, 0.048)
Duration of protection			
5 years (Ref)			
10 years	0.407 (0.031)	<0.001	(0.346, 0.467)
30 years	0.288 (0.029)	<0.001	(0.232, 0.345)
Out-of-pocket cost			
0–1000 CNY (Ref)			
1000–2000 CNY	0.007 (0.031)	0.811	(−0.053, 0.067)
2000–3000 CNY	−0.121 (0.029)	<0.001	(−0.178, −0.064)

SE: standard error.

**Table 5 vaccines-13-00906-t005:** Influence of vaccine attributes on vaccine choice preferences based on the mixed logit model (Model 2).

	Mean			SD	
Attributes and Levels	Coefficient	SE		Coefficient	SE
Protective efficacy					
80–90% (Ref)					
90–100%	−0.028	0.045		0.601 **	0.061
Duration of protection					
5 years (Ref)					
10 years	0.456 **	0.049		0.650 **	0.088
30 years	0.253 **	0.042		0.277 *	0.141
Out-of-pocket cost					
0–1000 CNY (Ref)					
1000–2000 CNY	0.040	0.036		−0.002	0.142
2000–3000 CNY	−0.179 **	0.051		0.506 **	0.087
Interaction terms					
Covariates × Protective efficacy					
Female × 90–100%	0.018	0.061		--	--
Educational level of over college × 90–100%	−0.039	0.064		--	--
Annual income of over 50,000 CNY × 90–100%	0.145	0.084		--	--
Covariates × Duration of protection					
Educational level of over college × 10 years	0.013	0.092		--	--
Educational level of over college × 30 years	0.083	0.081		--	--
Health workers × 10 years	0.224	0.134		--	--
Health workers × 30 years	0.397 **	0.122		--	--
Covariates × Out-of-pocket cost					
Female × 2000–3000 CNY	0.114	0.068		--	--
Annual income of over 50,000 CNY × 2000–3000 CNY	−0.107	0.089		--	--

SD: standard deviation. * indicates *p* < 0.05, ** indicates *p* < 0.001.

**Table 6 vaccines-13-00906-t006:** Calculation of willingness-to-pay based on mixed logit model.

Attributes	Willingness-to-Pay (WTP, CNY)	95% CI for WTP	
		Lower	Upper
Protective efficacy			
80–90%→90–100%	0.102	−0.695	0.900
Duration of protection			
5 years→10 years	7.077 **	3.511	10.643
5 years→30 years	4.840 **	2.280	7.399

** indicates *p* < 0.001.

**Table 7 vaccines-13-00906-t007:** Analysis of heterogeneity in hepatitis E vaccine choice preferences among residents of different places of residence.

Attributes and Levels	Rural (*n* = 2536)			Urban (*n* = 663)	
	Coefficient (SE)	*p*-Value		Coefficient (SE)	*p*-Value
ASC	−19.243 (451.073)	0.966		−18.990 (663.338)	0.977
Protective efficacy					
80–90% (Ref)					
90–100%	−0.010 (0.030)	0.739		0.106 (0.077)	0.168
Duration of protection					
5 years (Ref)					
10 years	0.449(0.044)	<0.001		0.722 (0.203)	<0.001
30 years	0.271 (0.037)	<0.001		0.707(0.103)	<0.001
Out-of-pocket cost					
0–1000 CNY (Ref)					
1000–2000 CNY	0.036(0.038)	0.345		0.042 (0.091)	0.644
2000–3000 CNY	−0.064 (0.037)	0.086		−0.511 (0.097)	<0.001

ASC: Alternative-Specific Constant, refers to the effect of the “opt-out” option in the model on the decision to vaccinate. Ref: Reference category; SE: Standard error; CNY: Chinese Yuan.

## Data Availability

The original contributions presented in this study are included in the article/Appendix A. Further inquiries can be directed to the corresponding authors.

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
