# Peer review of "Hepatitis E Vaccination Preferences and Willingness-to-Pay Among Residents: A Discrete Choice Experiment Analysis"

_vaccines, 2025, doi:10.3390/vaccines13090906_

Round 1

Reviewer 1 Report

Comments and Suggestions for Authors

Dear Editor,

The manuscript entitled “Hepatitis E Vaccination Preferences and Willingness-to-Pay among Residents: A Discrete Choice Experiment Analysis” addresses a relevant and timely public health issue, given the area of vaccine demand. The application of a discrete choice experiment (DCE) to assess vaccine preferences and willingness to pay (WTP) is appropriate and, in principle, aligns with established methodologies in health economics and behavioral epidemiology. Plese find below some major considerations:

L56 – Incidence rate values could be made available in the text for the reader.

L68 – “However, underdiagnosis due to suboptimal HEV detection sensitivity likely underestimates disease burden, underscoring the urgency for enhanced vaccination in high-risk populations and foodborne transmission control.” – Here, the authors should include some reference justifying the statement that there is underreporting of HEV cases in the region.

L77 – “Globally, systematic vaccination campaigns achieved 58% full vaccination coverage in South Sudan’s high-endemicity zones [13] and Bentiu region has reached 95% coverage in displaced persons camps (27,000 cases) by April 2022.” – This sentence seems out of context. Why the comparison with South Sudan? And Bentiu?

L80 – What do the authors mean by “effective containment outcomes.”? L71-L86 – The entire paragraph seems confusing, with information shifting between China, Sudan, and back to China.

L89 - New Zealand analyses identify adverse event risks as a public priority, while adolescents emphasize disease burden [17] – this sentence should be rewritten for better understanding.

Methods and Results

L101 – The authors should add a map of the study area, showing the cities and geographic regions sampled and listing the eight cities in the text (without using etc.)

L121 - The process of attribute and level selection (Section 2.2.1) is not sufficiently clear. It is unclear how expert validation was conducted—no information is provided on the number, profile, or consensus methods of the consulted experts. Furthermore, what's is the rationale for excluding two of the initial five attributes?. For example, the claim that the "severe adverse event" attribute had marginal utility is not accompanied by any quantitative evidence from pre-surveys. 

L134 - For the DCE framework, the authors mention using an orthogonal fractional factorial design but don´t specify the design algorithm used, the D-efficiency score, or how the nine profiles were constructed and paired into choice sets.

L166-179 - The minimum sample size for the DCE was estimated to be 150. However, the authors opted for an N of 3,300 participants. No statistical criteria were presented here, only assumptions based on the number of parameters evaluated. Furthermore, why was a 20% buffer suggested? "Based on the above comprehensive considerations, after rigorous calculations and adjustments, the final target sample size for this study was determined to be 3,300 participants." These "rigorous calculations" are unclear. I suggest that this paragraph be revised or deleted.

- On data collection, the authors do not specify how participants were identified. 

L193 - It is unclear which variables were included as covariates in the models. Furthermore, how the evaluation of the subgroup preference heterogeneity was carried out. the authors should provide a description of the modeling choices, estimation procedures, and validation steps.

-The manuscript applies both a conditional logit model and a mixed logit model, which is a common and methodologically sound approach in discrete choice experiment studies. However, the authors do not clearly explain why the mixed logit model was employed or how it improves upon the conditional logit. A purpose of the mixed logit would be to capture unobserved heterogeneity in preferences, but the manuscript provides no evidence that such heterogeneity exists or is statistically significant

-Shouldn't the authors need some statistics to demonstrate that the mixed logit model provides a better fit to the data? such as log-likelihood values, AIC?

Author Response

Comments 1: L56 – Incidence rate values could be made available in the text for the reader.
Response 1: We agree with this comment. Therefore, we added incidence data in lines 61-62 on page 2.

Comments 2: L68 – “However, underdiagnosis due to suboptimal HEV detection sensitivity likely underestimates disease burden, underscoring the urgency for enhanced vaccination in high-risk populations and foodborne transmission control.” – Here, the authors should include some reference justifying the statement that there is underreporting of HEV cases in the region.
Response 2: Thank you for pointing this out. We have added relevant references in line 76 on page 2.

Comments 3: L77 – “Globally, systematic vaccination campaigns achieved 58% full vaccination coverage in South Sudan’s high-endemicity zones [13] and Bentiu region has reached 95% coverage in displaced persons camps (27,000 cases) by April 2022.” – This sentence seems out of context. Why the comparison with South Sudan? And Bentiu?
Response 3: L90-This section mainly shows the vaccination status of hepatitis E vaccines worldwide. Although vaccination rates appear to be high in other countries, this is only true for specific populations, highlighting the low vaccination rate of this vaccine worldwide.

Comments 4: L80 – What do the authors mean by “effective containment outcomes.”? L71-L86 – The entire paragraph seems confusing, with information shifting between China, Sudan, and back to China.
Response 4: The two references mentioned in the article do not specifically mention the results achieved after vaccination, and the phrase “effective containment outcomes” is indeed inappropriate, so the sentence has been deleted. China was mentioned at the beginning of this section mainly to highlight that this hepatitis E vaccine was developed in China and is the only one available globally (in line 80-89 on page 2.). The WTO has recognized the vaccine's protective effect on the population, and therefore advocates its use worldwide. The subsequent mention of South Sudan, Bentiu, and China is mainly to illustrate the global vaccination situation (in line 90-94 on page 2.).

Comments 5: L89 - New Zealand analyses identify adverse event risks as a public priority, while adolescents emphasize disease burden [17] – this sentence should be rewritten for better understanding.
Response 5: Based on the reviewer's suggestion, this sentence has been rewritten (Page 3, lines 106 to 108).

Comments 6: L101 – The authors should add a map of the study area, showing the cities and geographic regions sampled and listing the eight cities in the text (without using etc.)Response 6: Thank you for pointing this out. We agree with this comment. We have added a map of the study area to Figure 1 (Page 4), which shows the sampling counties and geographical areas. The eight specific cities to which they belong are listed in the main text and on page 3, lines 138 to 139.

Comments 7: L121 - The process of attribute and level selection (Section 2.2.1) is not sufficiently clear. It is unclear how expert validation was conducted—no information is provided on the number, profile, or consensus methods of the consulted experts. Furthermore, what's is the rationale for excluding two of the initial five attributes? For example, the claim that the "severe adverse event" attribute had marginal utility is not accompanied by any quantitative evidence from pre-surveys.
Response 7: We have revised this section based on the reviewers' suggestions. Specific implementation methods verified by experts have been added to lines 187-237 of the text, including relevant information such as the number of participating experts and their backgrounds. Reasons for excluding two of the initial five attributes are also provided.

Comments 8: L134 - For the DCE framework, the authors mention using an orthogonal fractional factorial design but don´t specify the design algorithm used, the D-efficiency score, or how the nine profiles were constructed and paired into choice sets.
Response 8: We have provided a detailed explanation of the DCE design section in lines 241-249 on page 6, including the design method, the D-efficiency score, and how to construct and pair to form a choice set.

Comments 9: L166-179 - The minimum sample size for the DCE was estimated to be 150. However, the authors opted for an N of 3,300 participants. No statistical criteria were presented here, only assumptions based on the number of parameters evaluated. Furthermore, why was a 20% buffer suggested? "Based on the above comprehensive considerations, after rigorous calculations and adjustments, the final target sample size for this study was determined to be 3,300 participants." These "rigorous calculations" are unclear. I suggest that this paragraph be revised or deleted.
Response 9: Thank you for pointing this out. We have re-described the determination of sample size on page 7, lines 268-292.

Comments 10: - On data collection, the authors do not specify how participants were identified.
Response 10: The inclusion and exclusion criteria for study participants are clearly stated in “2.1. Participant Recruitment” located on page 7, lines 294-297.

Comments 11: L193 - It is unclear which variables were included as covariates in the models. Furthermore, how the evaluation of the subgroup preference heterogeneity was carried out. the authors should provide a description of the modeling choices, estimation procedures, and validation steps.
Response 11: Thank you for your insightful and constructive comments. The points you raised regarding the inclusion of model covariates, methods for assessing subgroup heterogeneity, and the lack of details on model selection and estimation are crucial for enhancing the rigor, reproducibility, and scientific value of this study. We fully agree on the necessity of these supplementary explanations and have made comprehensive and detailed additions in the revised manuscript: We have added Table 1 (page 8, line 334) in the ““2.5. Data Analysis Methods” section, which lists all the variables included in the study. Corresponding supplements have also been made regarding subgroup analysis and model selection and estimation (pages 8, lines 313-333).

Comments 12: -The manuscript applies both a conditional logit model and a mixed logit model, which is a common and methodologically sound approach in discrete choice experiment studies. However, the authors do not clearly explain why the mixed logit model was employed or how it improves upon the conditional logit. A purpose of the mixed logit would be to capture unobserved heterogeneity in preferences, but the manuscript provides no evidence that such heterogeneity exists or is statistically significant
Response 12: Thank you for pointing this out. We have presented the log-likelihood values, AIC, and BIC results for the two models in Table 3 (Page11). By comparing them, we can see that the mixed logit model performs better than the conditional logit model.

Comments 13: -Shouldn't the authors need some statistics to demonstrate that the mixed logit model provides a better fit to the data? such as log-likelihood values, AIC?
Response 13: Agree. We have, accordingly, made the necessary modifications to emphasize this point. Specifically, the log-likelihood values, AIC, and BIC for the conditional logit model and mixed logit model have been added to Table 3 (Page 11) to show the fitting effects of the two models.

Reviewer 2 Report

Comments and Suggestions for Authors

Dear Authors,
The paper is very interesting and relevant.
It is very well written and methodologically sound.
I would like to make just two small comments.
In Figure 2, in the range of 1,000–2,000 CNY, the beta coefficient is 0.169 and p = 0.011. However, the lower limit of the 95% confidence interval is very close to 0. This must be taken into account in the discussion of the results (lines 300–301). Although significant, the effect is very small.
On the other hand, the standard deviation is a measure of dispersion. It is not a measure of precision. The measure of precision is the standard error.
I propose using the mean (sd) or mean +/- SE.
You can read these papers.
1. Altman DG, Gore SM, Gardner MJ, Pocock SJ. Statistical guidelines for contributors to medical journals. Br Med J 1983;286:1,489-1,493
2. Bailar JC, Mosteller F. Guidelines for statistical reporting in articles for medical journals: amplifications and explanations. Ann Intern Med 1988;108:266-273
3. Tobías A.[Mean +/- SD, an incorrect expression].Med Clin (Barc). 1998 Feb 7;110(4):157

Author Response

Comments 1: In Figure 2, in the range of 1,000–2,000 CNY, the beta coefficient is 0.169 and p = 0.011. However, the lower limit of the 95% confidence interval is very close to 0. This must be taken into account in the discussion of the results (lines 300–301). Although significant, the effect is very small.
Response 1: Thank you for pointing this out. We agree with this comment. Therefore, we modified the wording and selected an indicator with clear statistical significance for description, specifically located on page 15, lines 464-465.

Comments 2: On the other hand, the standard deviation is a measure of dispersion. It is not a measure of precision. The measure of precision is the standard error. I propose using the mean (sd) or mean +/- SE. You can read these papers.
1. Altman DG, Gore SM, Gardner MJ, Pocock SJ. Statistical guidelines for contributors to medical journals. Br Med J 1983;286:1,489-1,493
2. Bailar JC, Mosteller F. Guidelines for statistical reporting in articles for medical journals: amplifications and explanations. Ann Intern Med 1988;108:266-273
3. Tobías A.[Mean +/- SD, an incorrect expression].Med Clin (Barc). 1998 Feb 7;110(4):157
Response 2: Agree. We have revised the wording in our paper and used the mean (sd) to represent age in Table 2 (Page 10).

Reviewer 3 Report

Comments and Suggestions for Authors
  1. It is stated that the sample was drawn from the 2017 seroepidemiological survey, yet the fieldwork was conducted in March–April 2024. It remains unclear how the sampling frame was updated or whether replacements were made to ensure contemporary representativeness. The authors should provide further details.
  2. A full factorial design of 18 profiles was employed, but 20 choice sets were subsequently generated using a D-optimal algorithm. It is inconsistent that this exceeds the theoretical total of 18 combinations, suggesting possible overlap or duplication of scenarios without adequate explanation.
  3. The Johnson–Orme formula (N > 500 c/(t·a)) is applied with t = 5 and a = 2, yet there is no description of how the blocks (four blocks of five sets) were accounted for. Moreover, the expansion to 3 300 appears arbitrary and is not quantified step by step, preventing verification of statistical power for subgroup analyses.
  4. The online–offline mixed mode may introduce access bias (excluding individuals without Internet access) and interviewer‐assisted bias; comparative analyses of responses by administration mode to assess measurement bias are not presented.
  5. Although a mixed logit model is mentioned, the specification of the random parameters (distributional assumptions, correlations, and whether variances were estimated) is not detailed. This omission hinders reproducibility and evaluation of whether heterogeneity was appropriately modeled.
  6. In the Methods section, heterogeneity is described as being explored via a conditional logit framework, yet subgroup estimates in the Results clearly derive from the mixed logit model (Table 4). It should be clarified which model—and under which assumptions—was actually used for each analysis.
  7. The WTP calculation assumes linearity of the cost parameter, whereas cost is presented in categorical ranges. The validity of computing βx/βcost when βcost originates from a discrete parameter is not discussed, which affects the monetary interpretation.
  8. Although efficacy (90–100% vs. 80–90%) is non-significant in both models (P = 0.915 and P = 0.768), the Discussion asserts that “the positive influence of efficacy… has been corroborated.” This contradicts the results and should be qualified or corrected.
  9. The sample comprises 79% rural respondents, yet provincial or national‐level recommendations do not account for contextual differences in provinces with varying socioeconomic structures. External validity should be discussed in greater depth before proposing broad policy implications.

Author Response

Comments 1: It is stated that the sample was drawn from the 2017 seroepidemiological survey, yet the fieldwork was conducted in March–April 2024. It remains unclear how the sampling frame was updated or whether replacements were made to ensure contemporary representativeness. The authors should provide further details.
Response 1: The 2017 seroepidemiological survey provided the prevalence rates of hepatitis E virus antibodies in different regions of Shaanxi Province, dividing the survey areas into “historically high HEV prevalence areas” and “historically low HEV
prevalence areas.” The 2024 study involved re-recruiting current (2024) residents within the “survey area” as study subjects. The individuals included in the 2024 survey are not the same as those from the 2017 survey.

Comments 2: A full factorial design of 18 profiles was employed, but 20 choice sets were subsequently generated using a D-optimal algorithm. It is inconsistent that this exceeds the theoretical total of 18 combinations, suggesting possible overlap or duplication of scenarios without adequate explanation.
Response 2: Agree. We have rewritten this section in detail, located in lines 241-249 on page 6, to facilitate reader comprehension.

Comments 3: The Johnson–Orme formula (N > 500 c/(t·a)) is applied with t = 5 and a = 2, yet there is no description of how the blocks (four blocks of five sets) were accounted for. Moreover, the expansion to 3 300 appears arbitrary and is not quantified step by step, preventing verification of statistical power for subgroup analyses.
Response 3: Thank you for pointing this out. We have re-described the determination of sample size on page 7, lines 268-292.

Comments 4: The online–offline mixed mode may introduce access bias (excluding individuals without Internet access) and interviewer-assisted bias; comparative analyses of responses by administration mode to assess measurement bias are not presented.Response 4: The wording in the “2.4. Data Collection” section may be inappropriate. In fact, participants were asked to scan a QR code using electronic devices or click on a questionnaire link (online) at the survey site (offline) to access the online questionnaire and ultimately complete it. The wording has been revised in lines 294-297 on page 7.

Comments 5: Although a mixed logit model is mentioned, the specification of the random parameters (distributional assumptions, correlations, and whether variances were estimated) is not detailed. This omission hinders reproducibility and evaluation of whether heterogeneity was appropriately modeled.
Response 5: Thank you for your valuable feedback. We fully agree on the necessity of clearly specifying distribution assumptions, correlations, and variance estimates for model reproducibility and heterogeneity assessment, and apologize for the omission in the original text. We have made the necessary revisions, setting the coefficients for protective efficacy, duration of protection, and out-of-pocket cost as random parameters, all of which follow a normal distribution. Additionally, we assume that these random parameters are mutually independent (lines 313–333 on page 8). The model estimates the standard deviation of all random parameters, as detailed in Table 5 (page 12).

Comments 6: In the Methods section, heterogeneity is described as being explored via a conditional logit framework, yet subgroup estimates in the Results clearly derive from the mixed logit model (Table 4). It should be clarified which model—and under which assumptions—was actually used for each analysis.
Response 6: We have revised the description in the Methods section on page 8, lines 313-333. Heterogeneity was explored using a mixed logit regression framework. This point is reiterated in the Results section (page 13, lines 401 and 414).

Comments 7: The WTP calculation assumes linearity of the cost parameter, whereas cost is presented in categorical ranges. The validity of computing βx/βcost when ββcost originates from a discrete parameter is not discussed, which affects the monetary interpretation.
Response 7: Thank you for pointing this out. We have noted the issue of cost data dispersion. In the study, categorical cost intervals were converted into continuous variables, primarily using the midpoint assignment method (e.g., the interval “1,000–2,000 CNY” was assigned a value of 1,500 CNY). This method has been applied in other analyses as well. To enhance the rigor of the article, we have added an explanatory paragraph in the "2.5. Data Analysis Methods" section (lines 339-344 on page 9) and clearly noted in the report of WTP results (lines 387-388 on page 12) that these results were calculated based on the converted continuous cost variable. An additional section discussing this issue has also been added to the discussion section (lines 506-516 on page 16).

Comments 8: Although efficacy (90–100% vs. 80–90%) is non-significant in both models (P = 0.915 and P = 0.768), the Discussion asserts that “the positive influence of efficacy… has been corroborated.” This contradicts the results and should be qualified or corrected.
Response 8: ““The positive influence of efficacy... has been corroborated.” This is a finding from other studies, which has been added to the sentence above (Page 15, lines 442). At the same time, we explain in lines 445-447 that this study did not observe similar results and the potential reasons for this.

Comments 9: The sample comprises 79% rural respondents, yet provincial or national level recommendations do not account for contextual differences in provinces with varying socioeconomic structures. External validity should be discussed in greater depth before proposing broad policy implications.
Response 9: Thank you for pointing out the key issues of sample representativeness and external validity. We fully agree that the 79% rural respondents in the sample and the lack of adequate coverage of provincial differences may affect the generalizability of the conclusions. Therefore, we have limited the geographical conditions of the recommendations in the conclusion section (lines 481-482 on page 16). Despite the coverage bias, the study has unique value in its in-depth exploration of the rural policy context.

Round 2

Reviewer 3 Report

Comments and Suggestions for Authors

Evaluate whether it is necessary to include references for the new texts included in the Introduction.

Author Response

Comments: Evaluate whether it is necessary to include references for the new texts
included in the Introduction.
Response: Thank you for pointing this out. We have added the relevant references to
the introduction on page 2, line 76. The previous references were added here to illustrate
that the insufficient sensitivity of hepatitis E virus detection methods may be one of the
reasons for the underestimation of the disease burden. However, one of these references
was published in 2016, and the other is from Mexico, which may raise issues of
practicality and applicability. Therefore, we have considered removing the previous two
references. Subsequently, additional references that better align with practicality and
requirements have been added.